# Towards Efficient Mixture of Experts: A Holistic Study of Compression Techniques

## Abstract

Scaling large language models has driven remarkable advancements across various domains, yet the continual increase in model size presents significant challenges for real-world deployment. The Mixture of Experts (MoE) architecture offers a promising solution by dynamically selecting and activating only a subset of experts during inference, thus substantially reducing computational costs while preserving high performance. Despite these benefits, MoE introduces new inefficiencies, such as excessive parameters and communication overhead. In this work, we present a holistic study on compression techniques of Mixture of Experts to enhance both efficiency and scalability. While recent efforts have focused on reducing the number of experts, these approaches still suffer from considerable communication and computational costs. To address this, we propose more aggressive strategies, such as Layer Drop, which removes entire MoE layers, and Block Drop, which eliminates transformer blocks. Surprisingly, these aggressive structure pruning techniques not only preserve model performance but also substantially improve efficiency. Additionally, beyond Expert Trimming, we also introduce Expert Slimming, which compresses individual experts to further boost performance and can be seamlessly integrated with Expert Trimming. Extensive experimental results demonstrate the effectiveness of our proposed methods — Layer Drop and Block Drop — along with the comprehensive recipe that integrates Expert Slimming and Expert Trimming, achieving a $6.05\times$ speedup with $77.1\%$ reduced memory usage while maintaining over $92\%$ of performance on Mixtral-$8\times$7B. Our code will be made publicly available upon acceptance.

## 1 Introduction

While scaling large language models has shown exceptional performance across various domains (Ramesh et al., 2021; OpenAI, 2024; Team, 2024a), the increasing model size poses significant challenges in real-world deployments (Sun et al., 2023; Frantar et al., 2022) due to excessive computational demands and associated costs. The Mixture of Experts (MoE) (Shazeer et al., 2017), which selectively activates a subset of parameters during inference, offers a promising solution to reduce these computational burdens. Additionally, integrating MoE with Large Language Models (LLMs) has been shown to enhance performance further (Jiang et al., 2024; Dai et al., 2024).

Despite these advances, MoE models still suffer from significant redundancies that increase deployment costs. tandard MoE implementations replicate feed-forward layers across multiple experts, resulting in models that are still heavily parameterized. For instance, Mixtral-$8\times$7B (Jiang et al., 2024) contains 47B parameters, but only 13B parameters are activated per token, leading to the substantial GPU memory consumption and limited scalability. In addition, replicating experts often introduces redundant experts. For example, He et al. (2023) observed that expert parameters could be compressed through parameter sharing. Similarly, Lu et al. (2024) noted that not all experts are essential, suggesting that some can be safely removed. These findings underscore the potential for compressing MoE models to improve efficiency without sacrificing effectiveness.

In this paper, we first investigate the Expert Trimming based compression techniques that reduce the number of experts to enhance the efficiency of MoE (Cheng et al., 2020; Liang et al., 2021). The most prevalent approach for Expert Trimming is Expert Drop, which scores each expert and drops the less important ones (Lu et al., 2024; Muzio et al., 2024). While Expert Drop reduces

model size, it does not eliminate the costly computations within the MoE layer and the complex communication among experts, leading to negligible improvements on the inference speed. To this end, we propose aggressive Expert Trimming methods to enhance MoE efficiency. Specifically, to mitigate communication and computation costs, we present Layer Drop that removes the entire MoE layer. Additionally, given the computation-intensive nature of the attention mechanism within transformer blocks, we further propose Block Drop, which removes the whole transformer blocks. We use similarity-based metrics to demonstrate the feasibility of Layer Drop and Block Drop. Surprisingly, these two coarse-grained methods outperform fine-grained Expert Drop by a large margin in balancing performance and efficiency. Additionally, with small-scale post-finetuning, the compressed models can be further optimized to achieve near-original performance.

Beyond removing experts, we also explore Expert Slimming, which focuses on compressing individual experts. Techniques such as network pruning (Han et al., 2016; Zhu & Gupta, 2017) and quantization (Jacob et al., 2017; Nagel et al., 2021) have proven effective for model compression, with quantization being particularly well-suited for hardware acceleration. By integrating Expert Slimming with Expert Trimming, we propose a unified framework for compressing MoE models that maximizes efficiency gains while maintaining strong performance.

Our experimental results on two widely-used MoE models, Mixtral-8×7B (Jiang et al., 2024) and DeepSeek-MoE-16B (Dai et al., 2024), demonstrate the effectiveness of our proposed methods. For Expert Trimming, Expert Drop significantly reduces the memory usage but it provides only marginal improvements in inference speed. In contrast, Layer Drop and Block Drop significantly accelerate inference and reduce memory usage while maintaining comparable performance to the original models. The combined strategy of Expert Trimming and Expert Slimming results in a 6.05× speedup with only 22.8% memory usage (20.0GB) while maintaining over 92% of the original performance on Mixtral-8×7B. The findings offer valuable insights for enhancing the efficiency of MoE models. Additionally, post-finetuning allows compressed models to recover most of their original performance, resulting in a minimal 0.6% performance gap compared to the uncompressed DeepSeek-MoE-16Bmodel.

In summary, by conducting a holistic study on compressing Mixture of Experts, our key contributions are as follows:

- We extend Expert Trimming to a higher architectural level by introducing Layer Drop and Block Drop, significantly improving the efficiency while maintaining the model performance.

- We introduce Expert Slimming, a method that compresses individual experts. By integrating Expert Slimming with Expert Trimming, we achieve further efficiency gains without compromising performance.

- Extensive experimental results demonstrate the effectiveness of our proposed methods, achieving a 6.05× speedup and reducing memory usage to just 20.0 GB, all while maintaining over 92% of performance on Mixtral-8×7B.

## 2    RELATED WORK

**Mixture of Experts**    The Mixture of Experts (MoE) is a kind of neural network architecture with an extended set of parameters (referred to as "experts") controlled by a router, which is first introduced in the context of conditional computation (Jacobs et al., 1991; Jordan & Jacobs, 1994). The potential of sparse activation in MoE is subsequently exploited by Shazeer et al. (2017) for efficient training and inference on pretrained models with special designs, opening the door for MoE in various vision (Riquelme et al., 2021) and language (Lepikhin et al., 2020; Du et al., 2022; Fedus et al., 2022) scenarios. Attributed to its exceptional efficiency, MoE has been adopted as a foundational framework in the designs of large language models (LLMs) (Jiang et al., 2024; Dai et al., 2024; Xue et al., 2024a; Zhu et al., 2024; Team, 2024b), achieving superior scaling laws at low computational costs (Clark et al., 2022). Further investigations emerge in developing improved expert structures (Gururangan et al., 2022; Rajbhandari et al., 2022; Dai et al., 2024), router designs (Lewis et al., 2021; Roller et al., 2021; Zhou et al., 2022), and training strategies (Shen et al., 2023; Chen et al., 2022), propelling the continuous evolution on the representation capability and computational efficiency of MoE models. Despite the success, MoE also suffers from efficiency issues. For instance, MoE replicates the experts,

significantly increasing the parameter budget (He et al., 2023). On the other hand, adopting multiple experts to process input tokens introduces communication costs and enhances latency (Song et al., 2023; Xue et al., 2024b).

**Compression Methods** The escalating size of large language models presents considerable hurdles for their practical implementation. Consequently, a range of efficient methods has emerged to address the implementation issues. Among them, model quantization (Frantar et al., 2022; Lin et al., 2024) and network pruning (Sun et al., 2023; Frantar & Alistarh, 2023) are widely utilized. Model quantization reduces the precision of neural network weights to lower bits (Jacob et al., 2017), while network pruning (Han et al., 2016) removes redundant parameters or architectures. Although these methods have shown promising results on dense models, they lack consideration for the inductive bias inherent in MoE. To bridge this gap, Expert Drop, as proposed in studies like (Muzio et al., 2024; Lu et al., 2024), addresses the unique nature of MoE by removing unimportant experts. By eliminating redundant experts, the MoE architecture becomes more compact and can be deployed at a lower cost. However, while Expert Drop leads to a more compact architecture, it may also lead to non-negligible performance drop and rely on post-training procedures for recovery.

## 3 PRELIMINARIES

### 3.1 MIXTURE OF EXPERTS

A Mixture of Experts (MoE) layer consists of a collection of $n$ experts, $\boldsymbol{E}_1, \boldsymbol{E}_2, \ldots, \boldsymbol{E}_n$, each associated with weights $\boldsymbol{W}_1, \boldsymbol{W}_2, \ldots, \boldsymbol{W}_n$, and a router $\boldsymbol{G}$ that dynamically selects the most relevant experts for a given input $\boldsymbol{x}$. The router computes selection scores, $\boldsymbol{G}(\boldsymbol{x}) \in \mathbb{R}^n$, for all experts and selects the top $k$ experts, resulting in a sparse activation pattern. The input $\boldsymbol{x}$ is processed by the selected experts, and their outputs are combined into a weighted sum based on the router's scores. This process is mathematically expressed as:

$$\mathcal{K} = \text{TopK}(\text{Softmax}(\boldsymbol{G}(\boldsymbol{x})), k), \tag{1}$$

$$\boldsymbol{y} = \sum_{i \in \mathcal{K}} \boldsymbol{G}(\boldsymbol{x})_i \cdot \boldsymbol{E}_i(\boldsymbol{x}|\boldsymbol{W}_i), \tag{2}$$

where $\mathcal{K}$ denotes the indices of selected experts, $\boldsymbol{G}(\boldsymbol{x})_i$ represents the selection score for the $i$-th expert, and $\boldsymbol{E}_i(\boldsymbol{x})$ is the output from the $i$-th expert. In transformer models, the MoE layer is often used as a replacement for the feed-forward network (FFN). In this context, each expert functions as an independent FFN module, enhancing the model's capacity without a proportional increase in the computational cost (Vaswani et al., 2017).

**Challenges** While MoE models have demonstrated strong performance across various tasks (Jiang et al., 2024; Dai et al., 2024), they also encounter significant deployment challenges. On one hand, MoE models replicate multiple expert networks, inflating model size and memory usage. For instance, Mixtral-8×7B has a total of 47B parameters, requiring 87.7GB of memory for deployment, though only 13B parameters are activated per token. On the other hand, the communication required to manage multiple expert networks increases latency and slows down inference speed, especially in distributed environments (Song et al., 2023; Yu et al., 2024).

### 3.2 OVERVIEW OF PREVIOUS COMPRESSION METHODS

To address the efficiency challenges, we first review several mainstream and state-of-the-art compression techniques for MoE models.

**Pruning**: Pruning reduces the number of active parameters by selectively disabling parts of the model's weights. In an MoE layer with $n$ experts $\boldsymbol{E}_i{i=1}^n$ and corresponding weights $\boldsymbol{W}_i{i=1}^n$, pruning introduces binary masks $\boldsymbol{M}_i{}_{i=1}^n$ to deactivate certain weights:

$$\hat{\boldsymbol{W}}_i = \boldsymbol{M}_i \odot \boldsymbol{W}_i. \tag{3}$$

Pruning can be *unstructured* (Lee et al., 2021; Bai et al., 2022), *semi-structured*, or *structured*. Unstructured sparsity tends to yield the best performance, semi-structured sparsity strikes a balance

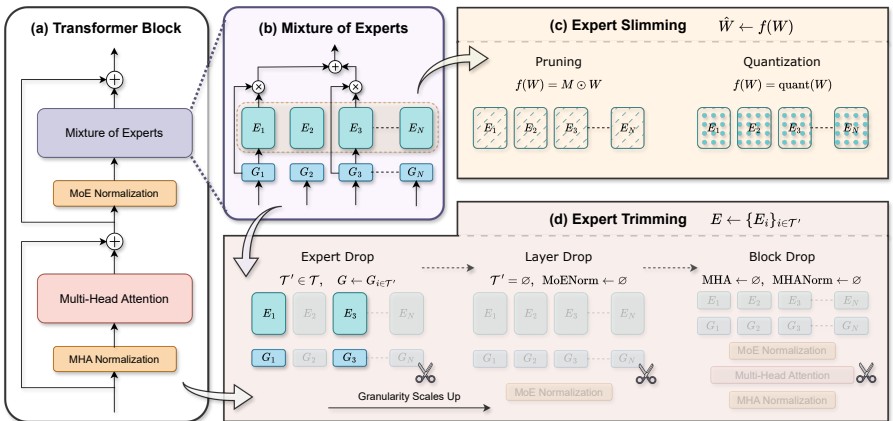

Figure 1: **The Unified View of MoE Compression.** The view integrates two complementary perspectives: Expert Slimming and Expert Trimming. Expert Slimming compresses individual experts, while Expert Trimming directly drops structured modules.

between efficiency and performance, and structured sparsity, while hardware-friendly, often results in lower performance.

**Quantization**: Unlike pruning, which involves masking out unimportant parameters, quantization reduces memory usage by converting model weights to lower-bit representations. For MoE layers, quantization is applied as follows:

$$\hat{\boldsymbol{W}}_i = \text{Quant}(W_i), \tag{4}$$

where "Quant" denotes the quantization function. Quantization decreases memory consumption without reducing FLOPs or the total number of parameters, making it particularly advantageous for hardware acceleration.

**Expert Drop**: Different from fine-grained pruning and quantization, Expert Drop entails the removal of expert networks, based on the observation that not all experts are equally important (Lu et al., 2024; Muzio et al., 2024). Given expert-wise importance scores $\boldsymbol{S}$ (e.g., the routing scores, $\boldsymbol{S}(\boldsymbol{E}_i) = \boldsymbol{G}(\boldsymbol{x})_i$), Expert Drop retains only the experts with the highest $n'$ scores:

$$\mathcal{T}' = \text{TopK}(\boldsymbol{S}(\{\boldsymbol{E}_i\}_{i=1}^n), n'), \tag{5}$$

$$\boldsymbol{E} \leftarrow \{\boldsymbol{E}_i\}_{i \in \mathcal{T}'}, \quad \boldsymbol{G} \leftarrow \boldsymbol{G}_{i \in \mathcal{T}'}. \tag{6}$$

Here, $\mathcal{T}'$ denotes the subset of the original expert indices $\mathcal{T} = \{1, 2, \ldots, n\}$. Expert Drop reduces FLOPs conditionally: when $\mathcal{T}'$ contains more than or equal to $k$ indices, MoE still utilizes the top $k$ experts for each input; otherwise, it uses all remaining experts. While this approach reduces communication between experts, the resulting speedup is usually insignificant when maintaining acceptable performance.

**Other Compression Techniques**: Other methods, such as low-rank decomposition (Li et al., 2024b;a), aim to compress model weights into smaller matrices, further reducing memory and computational costs. In this work, we primarily focus on the widely-used methods (pruning, quantization, and Expert Drop), leaving a more detailed exploration of these additional methods for future research.

## 4 A HOLISTIC STUDY OF MOE COMPRESSION TECHINIQUES

In this section, we propose a general framework that unifies various compression methods for MoE. This framework provides a comprehensive understanding of MoE model efficiency issues and helps identify new design spaces for further performance improvements.

### 4.1 OVERVIEW

Existing MoE compression methods primarily address two types of inefficiencies: **structural re-dundancies** in the overall architecture and **internal redundancies** within individual experts. To address both issues, we categorize these methods into two complementary perspectives: Expert

Table 1: **Summary of Compression Methods.** "✓" means effective and "✗" means ineffective, while "○" represents conditionally effective, depending on specific settings and environments.

| | Method | Formulation | Parameter | Memory | FLOPs | Speedup |
|---|---|---|---|---|---|---|
| Expert Trimming | Expert | $\mathcal{T} \leftarrow \mathcal{T}'$ | ✓ | ✓ | ○ | ○ |
| | Layer | | | | | |
| | Block | $\mathcal{T} \leftarrow \varnothing$ | ✓ | ✓ | ✓ | ✓ |
| Expert Slimming | Pruning | $\boldsymbol{M} \odot \boldsymbol{W}$ | ✓ | ○ | ✓ | ○ |
| | Quantization | $\mathrm{Quant}(\boldsymbol{W})$ | ✗ | ✓ | ✗ | ✓ |

Trimming focuses on removing structured components (e.g., experts, layers, or blocks), and Expert Slimming that compresses individual experts through techniques like pruning or quantization. An overview of these perspectives is illustrated in Figure 1.

Expert Trimming deals with compressing structured modules by selecting and retaining only a subset of the experts, denoted as $\mathcal{T}'$. This is represented by the transformation $\mathcal{T} \leftarrow \mathcal{T}'$. Methods like Expert Drop, which selectively drops unimportant experts, are examples of this approach. On the other hand, the compression of individual experts (Expert Slimming) focuses on the transformation and reduction of expert weights, denoted as $\boldsymbol{W}$. We utilize a transformation function $f(\boldsymbol{W})$ to represent this process. The transformation function $f(\boldsymbol{W})$ can be understood as a general mapping that applies various compression techniques to the weights of the model. For example, in pruning, $f(\boldsymbol{W})$ could be a function that sets a subset of the weights to zeros. In quantization, $f(\boldsymbol{W})$ might reduce the precision of the weights from 32-bit floats to 8-bit integers. By integrating these two perspectives, we can derive a general form for efficient MoE models. The compression **within** and **across** experts can be expressed as follows:

$$\boldsymbol{y} = \sum\nolimits_{i \in \mathcal{T}'} \boldsymbol{G}_i \cdot \boldsymbol{E}_i(\boldsymbol{x} | f(\boldsymbol{W}_i)). \tag{7}$$

In the following sections, we will elaborate on Expert Trimming and Expert Slimming, respectively.

## 4.2 EXPERT TRIMMING

The core operation of Expert Trimming involves updating the set of remaining experts denoted as $\mathcal{T} \leftarrow \mathcal{T}'$, where $\mathcal{T}'$ is a subset of the original expert indices $\mathcal{T}$. Specifically, Expert Drop updates the experts and their corresponding routing weights as follows: $\boldsymbol{E} \leftarrow \{\boldsymbol{E}_i\}_{i \in \mathcal{T}'}$ and $\boldsymbol{G} \leftarrow \boldsymbol{G}_{i \in \mathcal{T}'}$.

However, Expert Drop carries the risk of collapsing feature transformation. The absence of certain experts can lead to incorrect selections for given inputs, thereby degrading model performance (Chen et al., 2022). Additionally, partially reducing experts can disrupt routing patterns, negatively impacting the model's overall efficiency and effectiveness. Despite its benefits, Expert Drop still retains the costly computation within each expert and the complex communication between experts. These limitations highlight the need for further optimization of Expert Trimming to promote the efficiency. By systematically analyzing the redundancies and inefficiencies inherent in MoE models, we propose extending beyond expert-level optimizations to identify new design spaces for efficiency improvements.

We propose two novel techniques: **Layer Drop** and **Block Drop**. Layer Drop focuses on removing entire MoE layers, which significantly reduces both computation and communication overhead. Block Drop extends this concept by eliminating entire blocks, including attention layers and MoE layers, within transformer models. These advanced techniques aim to streamline the model architecture, improve performance, and enhance overall efficiency. -

Figure 2: **Illustration of Similarity Measurements in Layer Drop.** Features for calculating $\boldsymbol{S}^{(\mathrm{M})}$ and $\boldsymbol{S}^{(\mathrm{NM})}$ are colored with red and blue, respectively.

**Layer Drop** Inspired by Raposo et al. (2024); Elhoushi et al. (2024), we consider a special scenario of Expert Drop where all experts are dropped ($\mathcal{T} \leftarrow \mathcal{T}' = \varnothing$), effectively removing entire MoE layers. We refer to this approach as Layer Drop. To perform Layer Drop, we use a similarity-based metric where high

similarity indicates high redundancy in transformation. One straightforward metric is the cosine similarity between the input $\boldsymbol{x}$ and the output $\boldsymbol{y} = \mathrm{MoE}(\boldsymbol{x})$:

$$S^{(\mathrm{M})} = \frac{\boldsymbol{x} \cdot \boldsymbol{y}}{||\boldsymbol{x}||_2\, ||\boldsymbol{y}||_2}, \quad \text{where} \quad \boldsymbol{y} = \mathrm{MoE}(\boldsymbol{x}). \tag{8}$$

However, this metric alone does not adequately capture the impact of the MoE layer within the context of a transformer block, which includes a layer normalization module ("Norm") (Ba et al., 2016) and residual connections (He et al., 2015). To address this, we propose concurrently removing both the MoE and Norm layers. This approach ensures that the similarity metric more accurately reflects the combined functionality of these layers, allowing for a more precise identification of redundancy and a streamlined model architecture, as illustrated in Figure 2. By considering the similarity between the raw residual input and the aggregated output, we can better evaluate the necessity of the MoE layer in the overall architecture:

$$S^{(\mathrm{NM})} = \frac{\boldsymbol{x}' \cdot \boldsymbol{y}'}{||\boldsymbol{x}'||_2\, ||\boldsymbol{y}'||_2}, \quad \text{where} \quad \boldsymbol{y}' = \boldsymbol{x}' + \mathrm{MoE}(\mathrm{Norm}(\boldsymbol{x}')). \tag{9}$$

**Block Drop**  Within a transformer block, Layer Drop removes the MoE layers but retains the computation-costly attention layers (Ribar et al., 2024; Zhang et al., 2023). To address this issue, we further utilize the same similarity-based metrics to investigate whether the attention layer can be dropped without a significant performance drop. If feasible, this allows us to drop the entire block within MoE models, thus enhancing efficiency. We introduce Block Drop as an extension of Layer Drop, which also removes the attention layers. Specifically, for the $i$-th block, we assess its importance score by evaluating the similarity between its inputs $\boldsymbol{x}_l$ and outputs $\boldsymbol{y}_l$. Compared to Expert Drop, both Layer Drop and Block Drop focus on structures beyond expert level, with the potential to further enhance the efficiency of MoE models.

### 4.3 Expert Slimming

Given that employing multiple experts in MoE significantly escalates parameters and inference costs, Expert Slimming, stemming from single-model compression techniques, targets the compression of individual expert weights $\boldsymbol{W}$ exclusively. We denote any efficient transformation function as $f(\cdot)$, which encompasses pruning $\boldsymbol{M} \odot \boldsymbol{W}$ and quantization $\mathrm{Quant}(\boldsymbol{W})$. Through the application of such functions, we reduce the redundancy within each expert and create several light-weighted slim experts, thus improving their intrinsic efficiency. However, it is important to note that Expert Slimming primarily focuses on compressing individual experts without addressing the redundancy across multiple experts. For maximum efficiency gains, Expert Slimmingand Expert Trimming can be integrated to compress both individual experts and structured components. We summarize the efficiency contributions of all the discussed Expert Trimming and Expert Slimming methods in Table 1, highlighting the unique advantages of each approach.

## 5 Experiments on Expert Trimming

n this section, we evaluate the effectiveness of Expert Trimmingtechniques, starting with Expert Drop, and comparing it with our proposed methods, Layer Dropand Block Drop. Implementation details are provided in Appendix A.

**Expert Drop: Performance Degradation with Limited Efficiency Gains**  While experts are specific structures in MoE, not all experts hold equal significance. Figure 11 visualizes the distribution of expert-wise importance scores, highlighting this variability. To systematically drop experts at varying proportions, we conduct experiments using both layer-wise and global dropping approaches (see Appendix A.3). Given the importance of shared experts (Appendix E), we only dropped normal experts for DeepSeek-MoE-16B. Under both settings, Expert Drop causes consistent performance degradation. For example, dropping 25% of experts in Mixtral-8×7B results in a 23% performance drop on the MMLU task. The efficiency improvement from Expert Drop is also marginal. For instance, dropping 12.5% of experts results in less than a 1% speedup, despite significant performance losses. More experimental results are available in Appendix F.

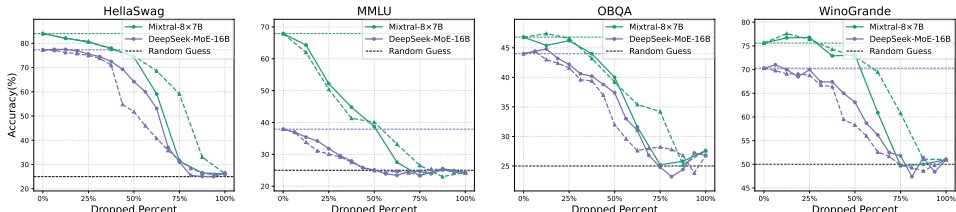

Figure 3: **Evaluation of Expert Drop.** We consider two strategies: layer-wise (dotted lines) and global (solid lines).

**Layer Drop: Comparable Performance with Greater Efficiency** To verify the feasibility of Layer Drop, we visualize feature similarity across different modules in Figure 4. This visualization shows a high level of similarity for features across the the MoE normalization module (Norm) and the MoE layer. In contrast, the low similarity for features across the MoE layer indicates the infeasibility of removing only MoE layers. Results from Figure 5 show that Layer Drop preserves performance within a wide range of compression ratio, e.g. $1\%$ performance drop on MMLU when dropping 8 layers for Mixtral-8×7B, revealing significant redundancy in the MoE layers.

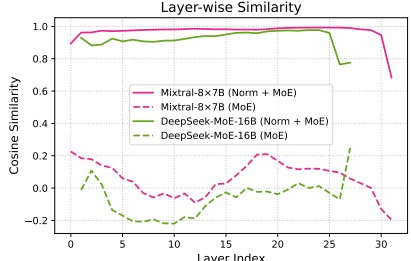

Figure 4: **Layer-Wise Similarity.** We consider two scenarios, i.e., for "MoE" and "Norm + MoE".

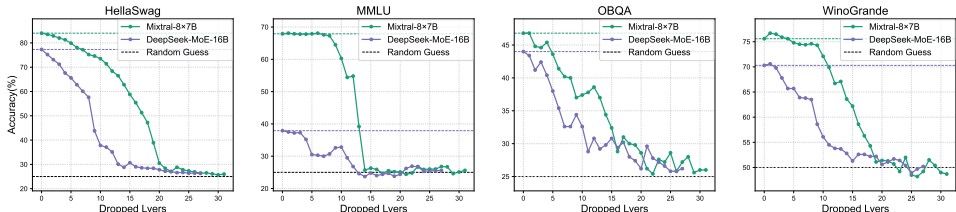

Figure 5: **Evaluation of Layer Drop.** We show results on Mixtral-8×7B and DeepSeek-MoE-16B (solid lines), along with the baseline and random guess performances (dotted lines).

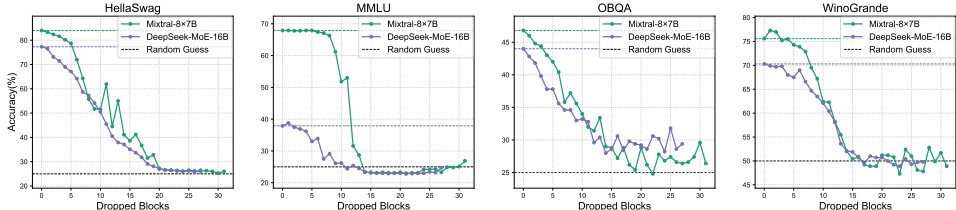

Figure 6: **Evaluation of Block Drop.** We show results on Mixtral-8×7B and DeepSeek-MoE-16B (solid lines), along with the baseline and random guess performances (dotted lines).

**Block Drop: Further Optimizing Efficiency by Pruning Entire Transformer Blocks** While Layer Drop maintains the performance of the original models, it still preserves the computation-costly attention layers. To address this, Block Dropextends Layer Dropby removing whole transformer blocks, including both MoE and attention layers, further reducing computational and memory costs. Figure 7 visualizes block-wise similarity, where both Mixtral-8×7B and DeepSeek-MoE-16B demonstrate high similarity between specific blocks. Based on this observation, we conduct the empirical study by varying the number of dropped blocks.

Surprisingly, as shown in Figure 6, the Mixtral-8×7B maintains over $90\%$ of the original performance even after removing 5 blocks (over 7 billion parameters). Similar observations are also found in DeepSeek-MoE-16B, where 4 blocks can be removed when maintaining $90\%$ performance. Since Block Drop removes computationally expensive attention layers, it outperforms Layer Drop by a large margin in terms of both memory and inference cost, as illustrated in Figure 8.

On the other hand, Block Drop prunes attention layers along with their corresponding KV-Cache Pope et al. (2022). For instance, an input sequence with a batch size of 128 and a sequence length of 2048 results in 32GB of KV-Cache, which can be reduced by 5GB using Block Drop. Overall, by targeting

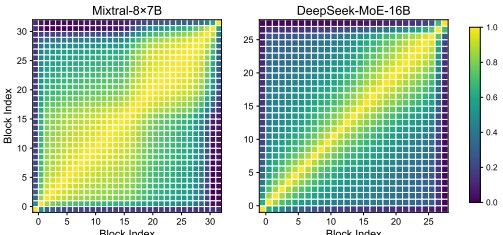 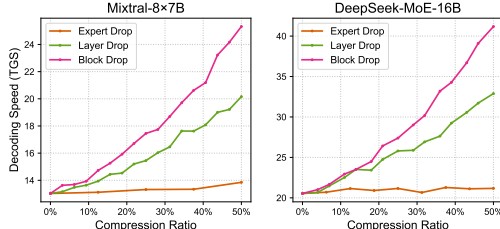

Figure 7: **Normalized Block-Wise Similarity.** We measure the cosine similarity among hidden features between blocks.

Figure 8: **Speedup Scaling Curves of Expert Trimming Methods.** where we measure the averaged decoding speed during generation.

higher-level structures, Layer Drop and Block Drop achieve substantial efficiency improvements while maintaining acceptable performance levels.

**MoE Layers are More Redundant than Dense Counterparts** Since Layer Drop and Block Drop can also be applied to dense models, we take Mistral-7B, the corresponding dense model of Mixtral-8×7B for comparison. Both models have the same depth and differ only in the FFN implementation, so we remove the same number of layers or blocks from each. When dropping an equal number of blocks, both MoE and dense models exhibit performance degradation. However, the MoE model suffers less performance drop under the same compression setting. For example, when dropping 8 MoE layers, the Mistral-7B receives a performance drop of 24.3, while Mixtral-8×7B only receives a drop of 7.0. This interesting finding highlights the higher redundancy in MoE layers, and further validates the effectiveness of applying Layer Drop and Block Drop to MoE models.

Table 2: **Comparison of Layer Drop and Block Drop on dense and MoE models.** "-L$n/m$", "-B$n/m$" represents dropping $n$ out of $m$ corresponding modules with Layer Drop and Block Drop, respectively.

| **Mistral-7B (Dense)** | | | | | |
|---|---|---|---|---|---|
| Method | ARC-C | HellaSwag | MMLU | OBQA | Average |
| Baseline | 61.5 | 83.7 | 62.5 | 43.8 | 62.9 |
| + L4/32 | 53.2 | 77.7 | 61.7 | 40.0 | 58.2 (-4.7) |
| + L8/32 | 36.7 | 33.6 | 53.3 | 30.6 | 38.6 (-24.3) |
| + B4/32 | 53.1 | 77.5 | 61.6 | 40.0 | 58.1 (-4.8) |
| + B8/32 | 40.0 | 63.9 | 60.0 | 30.6 | 48.6 (-14.3) |

| **Mixtral-8×7B (MoE)** | | | | | |
|---|---|---|---|---|---|
| Method | ARC-C | HellaSwag | MMLU | OBQA | Average |
| Baseline | 59.4 | 84.0 | 67.9 | 46.8 | 64.6 |
| + L4/32 | 56.2 | 81.3 | 67.6 | 44.6 | **62.4** (-2.2) |
| + L8/32 | 47.7 | 75.2 | 67.3 | 40.0 | **57.6** (-7.0) |
| + B4/32 | 53.8 | 80.2 | 67.9 | 43.0 | 61.2 (-3.4) |
| + B8/32 | 40.8 | 55.8 | 66.3 | 37.2 | 50.0 (-14.6) |

# 6 VISUALIZATION EXAMPLES OF LAYER DROP AND BLOCK DROP

In this section, we visualize the layer-wise similarity and the corresponding dropping order of MoE layers and blocks to investigate the varying levels of redundancy across different depths.

Since our similarity-based metrics depend on the hidden states of each block, the choice of data may influence feature similarity across layers. To investigate this, we conducted ablation studies on Mixtral-8×7B, examining both the number of samples and the types of datasets used for feature extraction. This analysis helps us understand how data selection affects decisions regarding the dropping of layers or blocks. The results are presented in Figure 9.

**Robustness to Calibration Datasets** In Figure 9a, we note that feature similarity remains relatively stable across different layers as the sample size increases, indicating that Layer Drop and Block Drop maintain consistency regardless of sample quantity. This confirms that using 128 samples suffices for computing similarity, which is adopted for all our experiments. Similarly, Figure 9b shows that varying the datasets, from pretraining with C4 to instruction tuning with Lima and MetaMathQA, does not significantly alter feature similarity. This demonstrates the resilience of Layer Drop and Block Drop to variations in data distribution.

**Redundant Deeper Layers** Figure 10 visualizes the remaining and dropped layers/blocks as the number of dropped modules increases. Both MoE architectures exhibit similar patterns in Layer Drop and Block Drop: initially, both models tend to drop the deeper layers, followed by the shallower

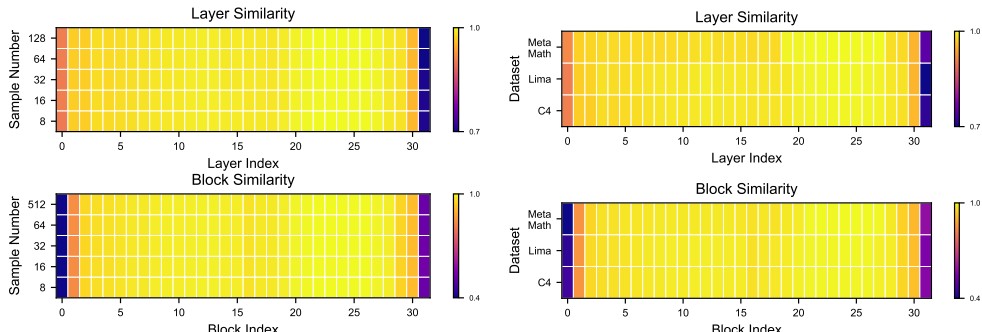

(a) Similarities under different number of samples. (b) Similarities under different datasets.

Figure 9: **Influence of Data Choices on Feature Similarity.** We measure the similarity among layers and blocks on Mixtral-8×7B. **(a)** The similarity calculated using different number of samples from C4 Raffel et al. (2019). **(b)** The normalized similarity calculated using $1,024$ samples from different datasets, i.e., C4, Lima Zhou et al. (2023) and MetaMathQA Yu et al. (2023).

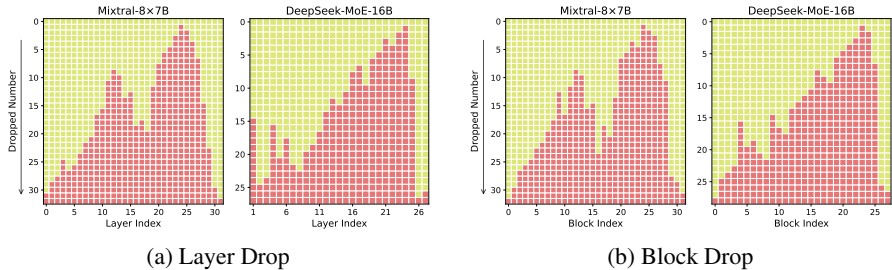

(a) Layer Drop (b) Block Drop

Figure 10: **Dropping Patterns for Layer Drop and Block Drop.** We visualize of the remaining layers and blocks under different dropped numbers, where yellow areas represent the retained portions and red areas indicate the dropped layers/blocks.

ones. These findings are consistent with Xu *et al.* Men et al. (2024), which suggests that deeper layers tend to be more redundant.

## 7 INTEGRATION OF EXPERT TRIMMING AND EXPERT SLIMMING

Beyond Expert Trimming, another avenue for MoE compression is Expert Slimming, which targets the compression of individual experts. Techniques such as quantization and network pruning are among the most commonly employed methods. We provide a detailed comparison of network pruning and quantization in Appendix B, where quantization outperforms in both performance and efficiency.

Since Expert Trimmingand Expert Slimmingfocus on different aspects of compression, we further explore their potential integration. Given the superior average performance and practical efficiency of quantization, we use it for Expert Slimming. For Expert Trimming, we include all three methods to offer a comprehensive comparison. The orders of applying these two compression techniques are discussed in Appendix D.

**Quantization Preserves the Performance of Expert Trimming**    As shown in Table 3, the integration of Expert Slimming and Expert Trimming significantly enhances overall efficiency. Quantization can be seamlessly combined with three different levels of dropping, achieving comparable performance. For instance, after quantization, the average performance of Layer Drop and Block Drop is nearly the same, maintaining more than $90\%$ of the performance of the original models.

**The Integration Significantly Enhances Efficiency**    In Table 3, the integration of Expert Trimming and quantization promotes efficiency by a large margin. Different Expert Trimming strategies showcase different advantages. Specifically, Expert Drop contributes to reducing memory usage but its speedup is marginal. Layer Drop and Block Drop excel in speedup as illustrated in Figure 8, with Block Drop demonstrating both higher performance and greater speedup. Considering all settings, the combination of Block Drop and quantization offers the best efficiency with comparable

Table 3: **Experimental Results of the Integration of Expert Trimming and Expert Slimming**. "-E$n/m$" denotes dropping $n$ out of $m$ experts per MoE layer on average. "-L$n/m$", "-B$n/m$" represents dropping $n$ out of $m$ layers/blocks with Layer Drop and Block Drop, respectively. The FLOPs are measured using an input with the $2,048$ sequence length.

| Mixtral-8×7B | | | | | | | | | | | |
|---|---|---|---|---|---|---|---|---|---|---|---|
| **Method** | SpeedUp | FLOPs | Memory | ARC-C | BoolQ | HellaSwag | MMLU | OBQA | PIQA | RTE | WinoGrande | Avg. |
| Baseline | – | 54.4T | 87.7GB | 59.4 | 84.2 | 84.0 | 67.9 | 46.8 | 83.8 | 70.4 | 75.6 | 71.5 |
| w/AWQ | 5.08× | 54.4T | 24.4GB | 58.4 | 84.2 | 83.3 | 66.6 | 45.8 | 83.0 | 69.0 | 76.3 | 70.8 |
| + E2/8 | 1.06× | 54.4T | 66.7GB | **53.2** | 77.7 | **80.5** | 52.2 | **46.2** | **81.7** | 55.6 | 76.8 | 65.5 |
| w/AWQ | 5.28× | 54.4T | **20.1GB** | 50.7 | 79.1 | **78.9** | 52.4 | 44.2 | 81.2 | 55.6 | 75.9 | 64.8 |
| + L8/32 | 1.19× | **42.9T** | 66.6GB | 47.7 | 85.3 | 75.2 | **67.3** | 40.0 | 75.8 | **69.7** | 74.6 | 67.0 |
| w/AWQ | 6.05× | **42.9T** | 20.0GB | 46.2 | 84.2 | 74.2 | **66.2** | 39.0 | 75.5 | 69.3 | 74.2 | 66.1 |
| + B5/32 | 1.17× | **46.0T** | 74.1GB | 51.3 | **85.3** | 78.7 | 67.9 | 42.0 | 79.3 | 69.7 | 74.3 | **68.6** |
| w/AWQ | 5.94× | **46.0T** | 21.9GB | 50.6 | **85.1** | 77.5 | 66.9 | 41.4 | 76.1 | **71.8** | 74.5 | **68.0** |

| DeepSeek-MoE-16B | | | | | | | | | | | |
|---|---|---|---|---|---|---|---|---|---|---|---|
| **Method** | SpeedUp | FLOPs | Memory | ARC-C | BoolQ | HellaSwag | MMLU | OBQA | PIQA | RTE | WinoGrande | Avg. |
| Baseline | – | 11.7T | 30.8GB | 48.1 | 72.4 | 77.3 | 37.9 | 44.0 | 80.4 | 63.9 | 70.3 | 61.8 |
| w/AWQ | 3.16× | 11.7T | 9.8GB | 46.8 | 71.2 | 76.6 | 36.4 | 43.6 | 80.1 | 62.1 | 70.1 | 60.9 |
| + E16/64 | 1.06× | 11.7T | 23.9GB | **45.0** | 67.1 | **75.6** | 31.8 | **42.2** | **80.2** | 59.9 | 70.0 | **59.0** |
| w/AWQ | 3.34× | 11.7T | **7.7GB** | **44.0** | 66.0 | **74.5** | 27.9 | **42.6** | **78.5** | 56.3 | 67.3 | 57.1 |
| + L4/28 | 1.14× | **10.6T** | 26.6GB | 39.5 | 70.2 | 67.6 | 35.2 | 40.4 | 75.8 | 48.4 | 65.7 | 55.3 |
| w/AWQ | 3.60× | **10.6T** | **8.5GB** | 42.1 | 72.0 | 69.2 | 33.7 | 39.8 | 75.1 | 47.7 | 66.5 | 55.8 |
| + B4/28 | 1.16× | **10.1T** | 26.4GB | 40.3 | 71.3 | 69.0 | **36.2** | 37.8 | 75.8 | 51.6 | 68.0 | 56.3 |
| w/AWQ | 3.67× | **10.1T** | **8.4GB** | 40.1 | 70.2 | 68.6 | **36.1** | 38.4 | 76.2 | 51.6 | 66.4 | 56.0 |

performance: a $6.05\times$ speedup with only 20.0GB memory usage, while maintaining over 92% of the performance on Mixtral-8×7B, making it available to be deployed on a NVIDIA RTX 3090 GPU.

# 8 POST-FINETUNING RECOVERS THE PERFORMANCE

While the discussed compression techniques maintains most of the performance of the original models, we further conduct post-finetuning to recover the degraded performance. Specifically, for comparison, we full-finetune DeepSeek-MoE-16B and corresponding compressed models on the Alpaca-GPT4 dataset Peng et al. (2023) for 3 epochs using a learning rate of 8e-6 with 0.03 warmup ratio and cosine scheduling, where the global batch size is set to 32. As shown in Figure 4, the post-finetuning process significantly reduces the performance gap between the compressed models and the original models, e.g. narrowing it from 5.5% to 0.6% for the model following Block Drop.

Table 4: **Performance of the DeepSeek-MoE-16B models finetuned after Expert Trimming**.

| DeepSeek-MoE-16B | | | | | | | | | | | |
|---|---|---|---|---|---|---|---|---|---|---|---|
| **Method** | SpeedUp | FLOPs | Memory | ARC-C | BoolQ | HellaSwag | MMLU | OBQA | PIQA | RTE | WinoGrande | Avg. |
| Baseline | – | 11.7T | 30.8GB | **48.1** | 72.4 | 77.3 | 37.9 | 44.0 | **80.4** | 63.9 | 70.3 | 61.8 |
| +SFT | | | | 44.6 | **75.3** | **79.0** | **40.3** | **44.6** | 80.3 | 70.4 | **71.7** | **63.3** |
| + E16/64 | 1.06× | 11.7T | 23.9GB | **45.0** | 67.1 | 75.6 | 31.8 | 42.2 | **80.2** | 59.9 | 70.0 | 59.0 |
| +SFT | | | | 44.4 | **74.0** | 78.6 | **38.5** | **45.8** | 79.6 | **65.7** | 70.1 | **62.1** |
| + L4/28 | 1.14× | 10.6T | 26.6GB | 39.5 | 70.2 | 67.6 | 35.2 | 40.4 | 75.8 | 48.4 | 65.7 | 55.3 |
| +SFT | | | | 42.1 | **78.9** | 75.2 | **40.8** | **43.4** | 77.6 | **71.1** | 69.5 | **62.3** |
| + B4/28 | 1.16× | 10.1T | 26.4GB | 40.3 | 71.3 | 69.0 | 36.2 | 37.8 | 75.8 | 51.6 | 68.0 | 56.3 |
| +SFT | | | | 43.2 | **78.2** | 75.0 | **40.4** | **43.8** | 76.8 | **74.0** | 70.2 | **62.7** |

# 9 CONCLUSION

In this paper, we conducted a holistic study of MoE compression techniques, facilitating a systematic understanding of the efficiency issue of MoE and identifying the new design space to improve the performance further. Based on this study, we propose a comprehensive recipe that integrates Expert Slimming and Expert Trimming to further enhance efficiency. Our proposed methods and insights not only address current challenges but also set the stage for future advancements in the field of MoE.

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

# A    IMPLEMENTATION DETAILS

## A.1    MODELS AND DATASETS

**Models.**    For our experiments, we employed Mixtral-8×7B Jiang et al. (2024) and DeepSeek-MoE-16B Dai et al. (2024). Mixtral-8×7B utilizes 8 experts for MoE layers and activates the top two for each input token. In contrast, DeepSeek-MoE-16B employs an dense FFN in the first block and utilizes two shared experts with additional 64 experts within MoE layers in other blocks.

**Datasets.**    For compression experiments, we used the C4 dataset Raffel et al. (2019), with 128 samples and an input sequence length of 2,048, following the setup in Sun et al. (2023); Lu et al. (2024); Lin et al. (2024); Frantar et al. (2022). To evaluate model performance, we report normalized zero-shot accuracy on the LM-harness benchmark, which includes multiple tasks: ARC-C Clark et al. (2018), BoolQ Clark et al. (2019), HellaSwag Zellers et al. (2019), MMLU Hendrycks et al. (2021), OBQA Mihaylov et al. (2018), PIQA Bisk et al. (2019), RTE Wang et al. (2019), and WinoGrande ai2 (2019). The evaluation code is based on EleutherAI LM Harness Gao et al. (2023).

## A.2    IMPLEMENTATION DETAILS OF EXPERT SLIMMING

Both Expert Slimming methods (i.e., pruning and quantization) require calibration data to estimate input statistics. To control this variable, we use 128 samples from the C4 dataset Raffel et al. (2019) as the calibration dataset for pruning. For quantization, we follow the default settings of GPTQ [1] and AWQ [2], using 128 random samples from Alpaca Taori et al. (2023) and Pile Gao et al. (2020), respectively. We use the default group size 128 for Mixtral-8×7B and 64 for DeepSeek-MoE-16B.

## A.3    IMPLEMENTATION DETAILS OF EXPERT DROP

The Expert Drop compresses MoE by preserving only important experts $\{\boldsymbol{E}_i\}_{i \in \mathcal{T}'}$ while removing others, where $\mathcal{T}'$ is determined by the importance scores $\{\boldsymbol{S}(\boldsymbol{E}_i)\}_{i \in \mathcal{T}}$. Following Muzio *et al.* Muzio et al. (2024), we measure the importance scores through the averaged routing scores of a batched data $\mathcal{X}$, i.e., $\{\boldsymbol{S}(\boldsymbol{E}_i)\} = \frac{1}{|\mathcal{X}|}\sum_{\boldsymbol{x} \in \mathcal{X}} \boldsymbol{G}_i(\boldsymbol{x})$, and consider two dropping strategies for Expert Drop: layer-wise dropping and global dropping.

**Layer-Wise dropping** removes the same number of experts for each layer. Given the total number of experts $n = |\mathcal{T}|$ and the preserved number of experts $n' = |\mathcal{T}'| < n$ in layer $l$, the preserved expert set $\mathcal{T}'^{(l)}$ is obtained by:

$$\mathcal{T}'^{(l)} = \{\boldsymbol{E}_t^{(l)}\}, \quad \text{where} \quad \boldsymbol{S}(\boldsymbol{E}_t^{(l)}) \in \text{TopK}(\{\boldsymbol{S}(\boldsymbol{E}_i^{(l)})\}_{i=1}^n, n'). \tag{10}$$

**Global dropping** constrains the total number of preserved experts for the entire model. Given the total number of layers $L$ in the model, the preserved expert set $\mathcal{T}'^{(l)}$ for layer $l$ is obtained by:

$$\mathcal{T}'^{(l)} = \{\boldsymbol{E}_t^{(l)}\}, \quad \text{where} \quad \boldsymbol{S}(\boldsymbol{E}_t^{(l)}) \in \text{TopK}\Big(\bigcup_{j=1}^m \{\boldsymbol{S}(\boldsymbol{E}_i^{(j)})\}_{i=1}^n, n'L\Big). \tag{11}$$

For the integration of Expert Slimming and Expert Trimming, we choose the global dropping as the strategy of Expert Drop, which shows competitive performance compared to the layer dropping for Mixtral-8×7B under low dropping ratios, as well as consistent better performance for DeepSeek-MoE-16B in Figure 13.

---

[1] https://github.com/AutoGPTQ/AutoGPTQ
[2] https://github.com/casper-hansen/AutoAWQ

# B  EXPERT SLIMMING

**Pruning: Comparable Performance with Deployment Challenges**  In Table 5, we evaluate representative pruning algorithms (i.e., Wanda Sun et al. (2023), SparseGPT Frantar & Alistarh (2023)) on Mixtral-8×7B and DeepSeek-MoE-16B. Since DeepSeek-MoE-16B utilizes both shared experts and normal experts, we conduct an ablation study on whether to prune shared experts, as discussed in Appendix E. We find that unstructured pruning preserves more than 95% of performance. However, it is not compatible with existing hardware. Conversely, the hardware-friendly semi-structured pruning (i.e., 4:8 and 2:4 patterns) undergoes a significant performance drop. Nevertheless, according to Lu *et al.* Lu et al. (2024), semi-structured sparsity is ineffective in speeding up MoE models.

Table 5: **Performance of Pruning on MoE.** We consider two mainstream pruning methods (i.e., Wanda Sun et al. (2023) and SparseGPT Frantar & Alistarh (2023)) under 50% unstructured sparsity and 2:4 semi-structured sparsity.

| Mixtral-8×7B | | | | | | | | | | |
|---|---|---|---|---|---|---|---|---|---|---|
| **Method** | Sparsity | ARC-C | BoolQ | HellaSwag | MMLU | OBQA | PIQA | RTE | WinoGrande | Avg. |
| Baseline | 0% | 59.4 | 84.2 | 84.0 | 67.9 | 46.8 | 83.8 | 70.4 | 75.6 | 71.5 |
| Wanda | 50% | 56.1 | 85.8 | 81.7 | 64.3 | 46.4 | 82.2 | 65.0 | 76.0 | 69.7 |
| SparseGPT | | 56.4 | 85.7 | 81.5 | 64.6 | 45.0 | 82.4 | 66.8 | 75.8 | 69.8 |
| Wanda | 2:4 | 51.4 | 79.4 | 77.8 | 60.3 | 44.0 | 80.7 | 65.3 | 74.1 | 66.6 |
| SparseGPT | | 49.2 | 81.0 | 77.6 | 59.2 | 44.0 | 80.6 | 63.9 | 74.8 | 66.3 |
| DeepSeek-MoE-16B | | | | | | | | | | |
| **Method** | Sparsity | ARC-C | BoolQ | HellaSwag | MMLU | OBQA | PIQA | RTE | WinoGrande | Avg. |
| Baseline | 0% | 48.1 | 72.4 | 77.3 | 37.9 | 44.0 | 80.4 | 63.9 | 70.3 | 61.8 |
| Wanda | 50% | 43.6 | 74.3 | 72.6 | 31.1 | 43.0 | 79.5 | 58.1 | 69.4 | 59.0 |
| SparseGPT | | 43.9 | 73.5 | 74.0 | 33.8 | 41.4 | 79.0 | 61.0 | 68.3 | 59.4 |
| Wanda | 2:4 | 38.2 | 66.1 | 67.5 | 27.6 | 39.4 | 77.0 | 53.8 | 66.7 | 54.5 |
| SparseGPT | | 43.1 | 68.9 | 71.6 | 27.6 | 41.6 | 78.3 | 57.4 | 66.6 | 56.9 |

**Quantization: Better Performance and Greater Efficiency**  In Table 6, we evaluate the impact of 4-bit quantization on MoE. Quantization offers two major benefits: it maintains the comparable performance of the original models and significantly reduces memory costs. Specifically, the quantized models achieve over 98% of the original performance while using less than 30% of the memory. Moreover, when quantized with AWQ Lin et al. (2024), Mixtral-8×7B and DeepSeek-MoE-16B achieve impressive speedups of ×5.08 and ×3.16, respectively. This demonstrates that 4-bit quantization is an effective technique for deploying MoE models in resource-constrained environments.

Table 6: **Performance of Quantization on MoE.** We utilize GPTQ Frantar et al. (2022) and AWQ Lin et al. (2024) as the quantization methods for 4-bit compression.

| Mixtral-8×7B | | | | | | | | | | | |
|---|---|---|---|---|---|---|---|---|---|---|---|
| **Method** | Bits | Memory | ARC-C | BoolQ | HellaSwag | MMLU | OBQA | PIQA | RTE | WinoGrande | Avg. |
| Baseline | 16 | 87.7GB | 59.4 | 84.2 | 84.0 | 67.9 | 46.8 | 83.8 | 70.4 | 75.6 | 71.5 |
| GPTQ | 4 | 24.4GB | 59.0 | 84.4 | 83.4 | 67.1 | 45.2 | 83.1 | 70.1 | 75.2 | 70.9 |
| AWQ | | | 58.4 | 84.2 | 83.3 | 66.6 | 45.8 | 83.0 | 69.0 | 76.3 | 70.8 |
| DeepSeek-MoE-16B | | | | | | | | | | | |
| **Method** | Bits | Memory | ARC-C | BoolQ | HellaSwag | MMLU | OBQA | PIQA | RTE | WinoGrande | Avg. |
| Baseline | 16 | 30.8GB | 48.1 | 72.4 | 77.3 | 37.9 | 44.0 | 80.4 | 63.9 | 70.3 | 61.8 |
| GPTQ | 4 | 9.8GB | 46.3 | 71.8 | 76.8 | 36.4 | 43.4 | 80.0 | 63.9 | 70.2 | 61.1 |
| AWQ | | | 46.8 | 71.2 | 76.6 | 36.4 | 43.6 | 80.1 | 62.1 | 70.1 | 60.9 |

## C  ANALYSIS ON THE DROPPING PATTERNS OF EXPERT DROP

**Score Distribution Directs Expert Drop.** The distribution of importance scores is informative to determine the proportion of dropped experts. In Figure 11, we visualize the score distribution of Expert Drop for Mixtral-8×7B and DeepSeek-MoE-16B, respectively. DeepSeek-MoE-16B, which allocates more experts, shows a left-skewed distribution where most experts have low scores. In contrast, Mixtral-8×7B demonstrates a right-skewed distribution, with only a few experts being deemed unimportant. This distribution difference results in different resistance capability against Expert Drop, where DeepSeek-MoE-16B can drop much more experts than Mixtral-8×7B while maintaining competitive performance, as demonstrated in Table 3 and Figure 13.

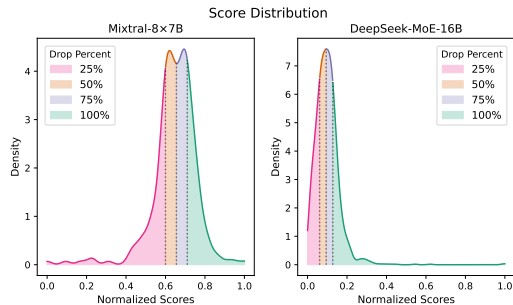

Figure 11: **Distribution of Normalized Importance Scores $S$ for Expert Drop.** We highlight the density of scores under different drop ratios with different colors.

**Global Expert Drop Removes Experts Fine-Grainedly.** We employed two different strategies for Expert Drop, namely layer-wise and global. Layer-wise dropping treats each layer equally by dropping the same number of experts, while global dropping results in different proportions of remaining experts across layers. We visualize the distribution of remaining experts after global dropping in Figure 12. We find the global dropping shows a more fine-grained pattern on dropping experts, where the bottom layers are more vulnerable under lower dropping ratios (yellow part).

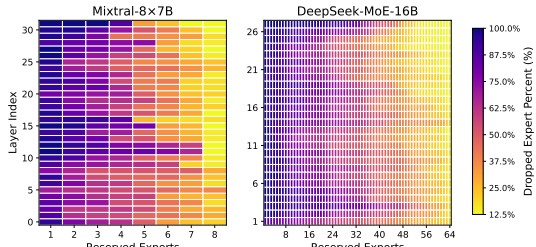

Figure 12: **Distribution of Dropped Experts for Expert Drop.** We visualize of the dropped experts under different drop ratios, where the dropped experts are colored from yellow to blue as the drop ratio increases.

# D    ABLATION STUDY ON COMPRESSION ORDERS

In Section 7, we discussed the combination of Expert Trimming and Expert Slimming. Here we ablate on the orders of compression when combining these two techniques. Results in Table 7 show that the order of Expert Trimming and Expert Slimming doesn't have a significant influence on the performance, where applying Expert Slimming then Expert Trimming ("S+T") performs slightly better for Mixtral-8×7B (e.g. +0.5, +0.4 and +0.1 for Expert Drop, Layer Drop and Block Drop, respectively). To this end, we choose "S+T" as the final implementation in our experiments.

Table 7: **Ablation results on different orders of Expert Slimming and Expert Trimming.** "S+T" denotes first applying Expert Slimming then Expert Trimming, and "T+S" denotes the reversed order.

| Mixtral-8×7B | | | | | | | | | |
|---|---|---|---|---|---|---|---|---|---|
| **Method** | ARC-C | BoolQ | HellaSwag | MMLU | OBQA | PIQA | RTE | WinoGrande | Avg. |
| Baseline | 59.4 | 84.2 | 84.0 | 67.9 | 46.8 | 83.8 | 70.4 | 75.6 | 71.5 |
| + E2/8, AWQ (S+T) | 50.7 | 79.1 | 78.9 | 52.4 | 44.2 | 81.2 | 55.6 | 75.9 | 64.8 |
| + E2/8, AWQ (T+S) | 50.8 | 79.9 | 78.7 | 49.2 | 44.4 | 80.9 | 55.2 | 75.4 | 64.3 |
| + L8/32, AWQ (S+T) | 46.2 | 84.2 | 74.2 | 66.2 | 39.0 | 75.5 | 69.3 | 74.2 | 66.1 |
| + L8/32, AWQ (T+S) | 46.8 | 84.4 | 74.0 | 65.3 | 39.8 | 75.0 | 66.8 | 73.2 | 65.7 |
| + B5/32, AWQ (S+T) | 50.6 | 85.1 | 77.5 | 66.9 | 41.4 | 76.1 | 71.8 | 74.5 | 68.0 |
| + B5/32, AWQ (T+S) | 50.3 | 84.7 | 77.4 | 65.8 | 42.0 | 78.8 | 70.4 | 74.0 | 67.9 |
| DeepSeek-MoE-16B | | | | | | | | | |
| **Method** | ARC-C | BoolQ | HellaSwag | MMLU | OBQA | PIQA | RTE | WinoGrande | Avg. |
| Baseline | 48.1 | 72.4 | 77.3 | 37.9 | 44.0 | 80.4 | 63.9 | 70.3 | 61.8 |
| + E16/64, AWQ (S+T) | 44.0 | 66.0 | 74.5 | 27.9 | 42.6 | 78.5 | 56.3 | 67.3 | 57.1 |
| + E16/64, AWQ (T+S) | 44.7 | 64.1 | 74.0 | 29.0 | 42.6 | 79.9 | 54.2 | 68.4 | 57.1 |
| + L4/28, AWQ (S+T) | 42.1 | 72.0 | 69.2 | 33.7 | 39.8 | 75.1 | 47.7 | 66.5 | 55.8 |
| + L4/28, AWQ (T+S) | 42.4 | 71.7 | 69.1 | 33.4 | 40.1 | 74.8 | 47.6 | 66.2 | 55.7 |
| + B4/28, AWQ (S+T) | 40.1 | 70.2 | 68.6 | 36.1 | 38.4 | 76.2 | 51.6 | 66.4 | 56.0 |
| + B4/28, AWQ (T+S) | 41.6 | 69.4 | 69.1 | 35.8 | 38.6 | 76.2 | 50.9 | 67.0 | 56.1 |

# E ABLATION STUDY ON SHARED EXPERTS IN DEEPSEEK-MOE-16B

While most MoE models follow Equation 2 to implement the experts, models like DeepSeek-MoE-16B adopt a residual Rajbhandari et al. (2022) form of experts, which brings a special scenario to discuss. In the residual MoE, an extra set of $m$ shared experts $\{\bar{E}_1, \bar{E}_2, \ldots, \bar{E}_m\}$ are always selected by the router $G$ and activated for all inputs. Given an input $x$, the output can be represented as a degenerated form of Equation 2, where the scores of shared experts are fixed to 1:

$$y = \sum_{i \in \mathcal{K}} G(x)_i \cdot E_i(x) + \sum_{j=1}^{m} \bar{E}_j(x). \tag{12}$$

This special form of expert routing may bring a difference in the redundancy distribution of MoE. Here we discuss the influence of shared experts through pruning and present the results in Table 8. We find that pruning without the shared experts will boost the performance at a considerable scale, i.e., $+3.6\%$ and $+1.5\%$ of the averaged accuracy for unstructured pruning with Wanda and SparseGPT, respectively. This finding reveals a different pattern of the inner redundancy in that the shared experts are less compressible compared to the others in residual MoE models, which may inform future work.

Table 8: **Ablation Study of Pruning Shared Experts on DeepSeek-MoE-16B.** We consider two scenarios, i.e., pruning both shared experts and normal experts ("w/Pruning Shared Experts") and pruning normal experts only ("w/o Pruning Shared Experts"). We use two mainstream pruning methods (i.e., Wanda Sun et al. (2023) and SparseGPT Frantar & Alistarh (2023)) under both unstructured sparsity (50%) and semi-structured sparsity (2:4).

| Method | Sparsity | ARC-C | BoolQ | HellaSwag | MMLU | OBQA | PIQA | RTE | WinoGrande | Avg. |
|--------|----------|-------|-------|-----------|------|------|------|-----|------------|------|
| **DeepSeek-MoE-16B** | | | | | | | | | | |
| Baseline | 0% | 48.1 | 72.4 | 77.3 | 37.9 | 44.0 | 80.4 | 63.9 | 70.3 | 61.8 |
| *w/ Pruning Shared Experts* | | | | | | | | | | |
| Wanda | 50% | 43.6 | 74.3 | 72.6 | 31.1 | 43.0 | 79.5 | 58.1 | 69.4 | 59.0 |
| SparseGPT | | 43.9 | 73.5 | 74.0 | 33.8 | 41.4 | 79.0 | 61.0 | 68.3 | 59.4 |
| Wanda | 2:4 | 38.2 | 66.1 | 67.5 | 27.6 | 39.4 | 77.0 | 53.8 | 66.7 | 54.5 |
| SparseGPT | | 43.1 | 68.9 | 71.6 | 27.6 | 41.6 | 78.3 | 57.4 | 66.6 | 56.9 |
| *w/o Pruning Shared Experts* | | | | | | | | | | |
| Wanda | 50% | 44.0 | 76.3 | 73.5 | 36.2 | 41.0 | 79.3 | 59.9 | 70.2 | 60.0 |
| SparseGPT | | 45.0 | 75.5 | 74.4 | 36.3 | 41.0 | 79.4 | 64.3 | 69.3 | 60.7 |
| Wanda | 2:4 | 40.1 | 75.7 | 69.9 | 33.5 | 40.0 | 77.9 | 58.8 | 68.6 | 58.1 |
| SparseGPT | | 40.7 | 75.7 | 69.9 | 33.3 | 39.0 | 77.7 | 61.4 | 69.4 | 58.4 |

# F    FULL EXPERIMENTAL RESULTS

We provide the full results of Expert Trimming, including Expert Drop, Layer Drop and Block Drop, in Figure 13, 14, and 15, respectively.

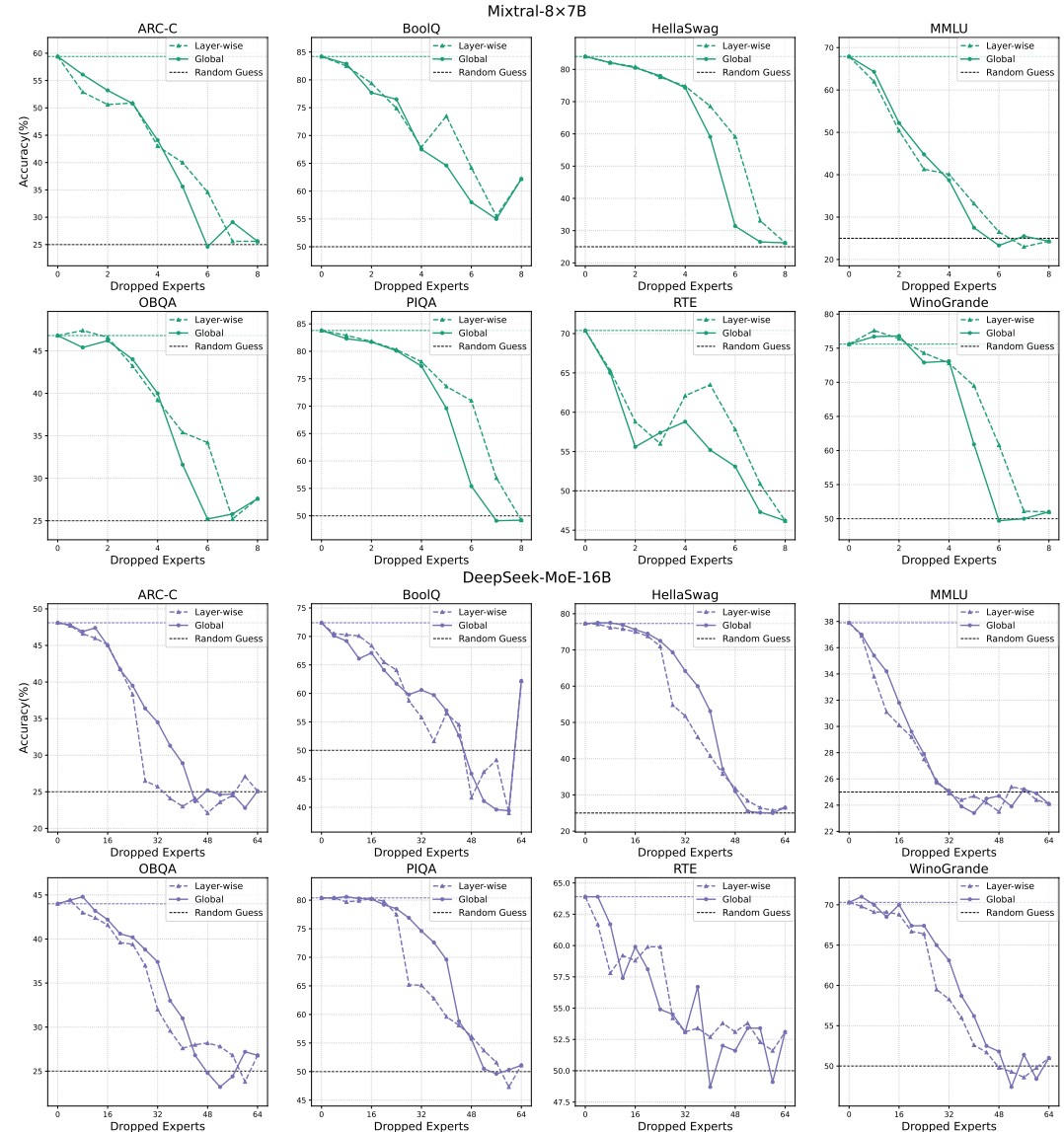

Figure 13: **Full Results for Expert Drop.** We consider two strategies: layer-wise (dotted lines) and global (solid lines).

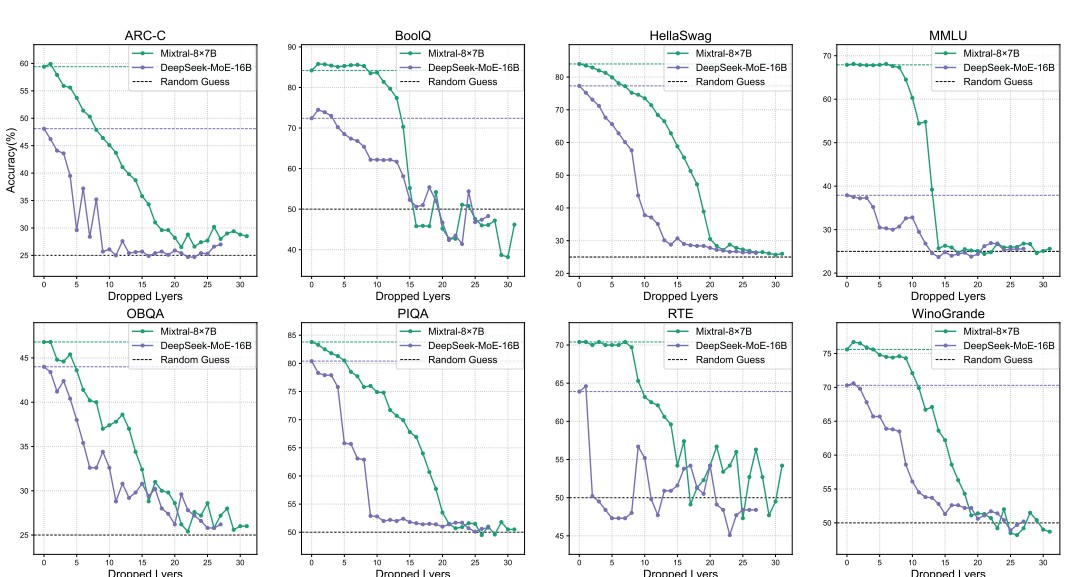

Figure 14: **Full Results for Layer Drop.** We show results on Mixtral-8×7B and DeepSeek-MoE-16B (solid lines), along with the baseline and random guess performances (dotted lines).

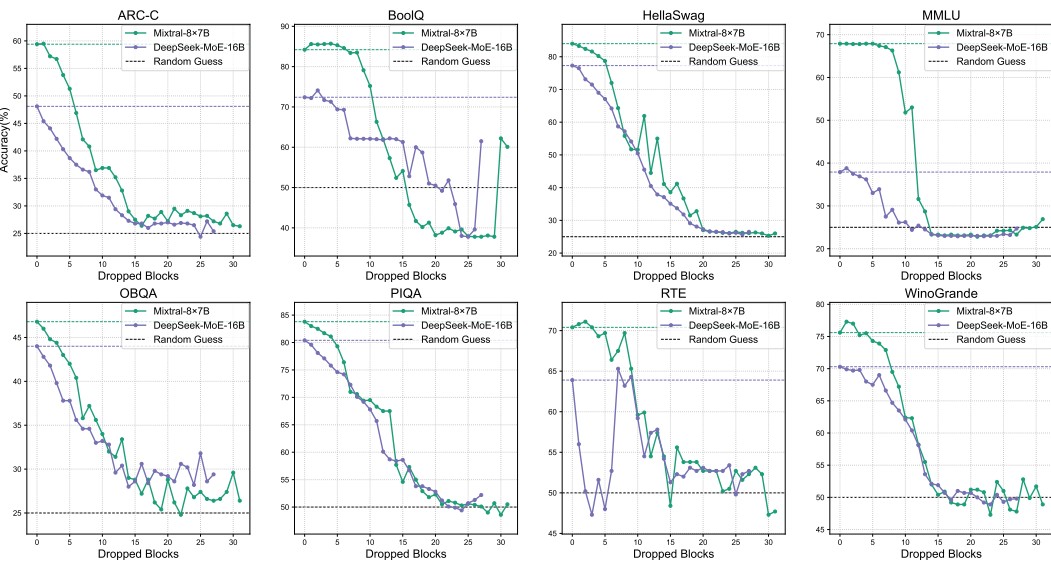

Figure 15: **Full Results for Block Drop.** We show results on Mixtral-8×7B and DeepSeek-MoE-16B (solid lines), along with the baseline and random guess performances (dotted lines).