# OpenReview forum: "Towards Efficient Mixture of Experts: A Holistic Study of Compression Techniques"
_ICLR.cc/2025/Conference — ICLR 2025 Conference Withdrawn Submission_

### Official Review · Reviewer_jCFm · 2024-10-28

**Soundness:** 3
**Presentation:** 3
**Contribution:** 2
**Rating:** 3
**Confidence:** 4

**Summary:**

This paper propose a pruning method for MoE LLMs including experts trimming: layer drop and block drop and Expert slimming: other parameter pruning or quantization strategy. Experiments on some LLM benchmarks are provided.

**Strengths:**

LLMs efficiency is an very important topic for the deployment of large language models. The idea proposed in this paper is simple.
The paper is clearly written.

**Weaknesses:**

1. Some typos:
(1) line 304: Expert Slimmingand -> Expert Slimming and
(2) line 311: n this Section -> In this Section
(3) line 311: Expert Trimmingtechniques -> Expert Trimming techniques
(4) line 43:  tandard MoE -> Standard MoE
(5) line264: efficiency. - -. efficiency.

2. The primary issue seems to be the paper's contributions. Expert Trimming, as implemented, is essentially layer pruning for LLMs, whether using layer drop or block drop techniques. Additionally, expert slimming does not appear to be a novel contribution here. Given this, the paper's contributions feel somewhat limited in scope.

3. Lack of comparison of important baselines. I believe there are many layer pruning baseline methods, It is important to give some comparisons in experiments.

**Questions:**

1. In line 292, "we assess its importance score by evaluating the similarity between its inputs x_l and outputs y_l", are x and y here similar as the x' and y' not x and y in layer drop?

2. Regarding Figure 3, it’s intriguing that the layer-wise expert drop consistently surpasses global strategies at high drop percentages. Could there be any underlying mechanism or explanation for why this approach maintains performance better?

3. On the training pipeline for LLMs, the typical stages are: (1) pretraining, (2) supervised fine-tuning (SFT), and (3) alignment. Based on the authors' findings, in which of these stages might the proposed pruning method be most effectively implemented?

4. This proposed expert or layer pruning method utilizes static pruning. Are there any reasons why dynamic pruning, which adjusts layers for different tokens, was not considered? or any comparisons?

5. From Table 3, there appears to be a performance discrepancy between Mistral MoE and DeepSeek MoE. Specifically, block drop or layer drop methods seem to perform considerably worse on DeepSeek. Could there be an underlying reason for this difference? So I am very confuse of the authors' conclusion that "in Layer Drop and Block Drop excel in speedup as illustrated in Figure 8, with Block Drop demonstrating both higher performance and greater speedup".

---

### Official Review · Reviewer_EyJP · 2024-10-31

**Soundness:** 2
**Presentation:** 3
**Contribution:** 2
**Rating:** 5
**Confidence:** 4

**Summary:**

This work investigates compression techniques for MoE, including Layer Drop and Block Drop, which remove MoE layers and transformer blocks, respectively, enhancing efficiency without compromising performance.

**Strengths:**

1. The paper is well-written.
2. The figures are easy to understand and visually appealing.
3. The observations related to expert trimming are meaningful, and the trade-offs between expert drop, layer drop, and block drop are well-considered.

**Weaknesses:**

1. The approach presented is straightforward, combining expert/layer/block drop with quantization to compress the MoE model. While the authors claim to introduce two novel techniques, Layer Drop and Block Drop, these methods are not truly new [1].
2. In line 371, the authors state, "Mixtral-8×7B maintains over 90% of the original performance"; however, would training from scratch with a similarly compact architecture also achieve comparable accuracy? Authors could conduct an ablation study comparing their compressed model to a model trained from scratch with a similar architecture.
3. Additionally, the authors propose simultaneously removing both the MoE and Norm layers, yet this strategy lacks sufficient novelty or justification. Authors may conduct ablation studies showing the impact of removing each layer type individually vs. together.
4. There are also typos in lines 311 and 312: “In this section, we evaluate the effectiveness of Expert Trimming techniques." Additionally, there's an extra "-" in line 264.
5. Regarding the results, Table 3 indicates that the combined use of Expert Trimming and Expert Slimming (quantization) provides advantages over using only AWQ. However, the performance improvement is marginal. Please discuss the practical significance of these marginal improvements.
6. There is no clear conclusion about which expert trimming is the best. The authors can include a more discussion. This could help readers quickly understand the trade-offs between the different approaches in terms of performance, efficiency, and ease of implementation.

[1] Reducing transformer depth on demand with structured dropout.

**Questions:**

N/A

---

### Official Review · Reviewer_Fvk3 · 2024-11-03

**Soundness:** 3
**Presentation:** 3
**Contribution:** 1
**Rating:** 5
**Confidence:** 3

**Summary:**

This paper presents a comprehensive study on compression techniques for Mixture of Experts (MoE) architectures to enhance efficiency and scalability in large language models. The authors propose Expert Trimming strategies like Layer Drop and Block Drop, which remove entire MoE layers and transformer blocks, and Expert Slimming to compress individual experts, all aimed at reducing computational costs and memory usage while preserving high performance. Experimental results demonstrate that these methods achieve a 6.05× speedup and 77.1% reduced memory usage, maintaining over 92% of the performance on the Mixtral-8×7B model, offering a promising solution for real-world deployment challenges.

**Strengths:**

Pros.
1.	Expert Trimming and Expert Slimming are introduced in the paper and are well-studied comprehensively.
2.	The evaluation result of Expert Trimming analyzes the impact of different compression methods, including Expert Drop, Layer Drop, and Block Drop. The analysis provides the instructive meaning to choose the compression method of Expert Trimming.
3.	The comprehensive compression performance comprised of Expert Trimming and Expert Slimming is provided.

**Weaknesses:**

Cons.
1.	The novelty is a big concern. The proposed method is a combination of existing works.
2.	The baselines of SOTA works are expected in the table to show the effectiveness of the proposed methods.

**Questions:**

See weeknesses

---

### Comment · Area_Chair_iN35 · 2024-11-24

Dear Reviewers,

This is a friendly reminder that the discussion period will end on Nov 26th (Anywhere on Earth). If you have not already, please take a careful look at the other reviews and author responses, and comment on whether your original rating stands. Thank you.

Best, AC

---

### Comment · Area_Chair_iN35 · 2024-11-28

Dear reviewers,

This is a friendly reminder that the discussion period has been extended until December 2nd. If you haven’t yet, we kindly encourage you to review the authors' rebuttal and messages at your earliest convenience and confirm whether your comments have been adequately addressed.

We greatly appreciate your service to this process.

Best, AC

---

### Note · Authors · 2024-11-30

I have read and agree with the venue's withdrawal policy on behalf of myself and my co-authors.